# Past HIV-1 Medications and the Current Status of Combined Antiretroviral Therapy Options for HIV-1 Patients

**DOI:** 10.3390/pharmaceutics13111798

**Published:** 2021-10-27

**Authors:** Matthew Weichseldorfer, Marvin Reitz, Olga S. Latinovic

**Affiliations:** 1Institute of Human Virology, School of Medicine, University of Maryland, Baltimore, MD 21201, USA; mweichseldorfer@ihv.umaryland.edu; 2Department of Medicine, School of Medicine, University of Maryland, Baltimore, MD 21201, USA; MReitz@ihv.umaryland.edu; 3Department of Microbiology and Immunology, School of Medicine, University of Maryland, Baltimore, MD 21201, USA

**Keywords:** HIV-1, AIDS, cART, entry inhibitors, LRA, reverse transcriptase inhibitors, protease inhibitors, integrase inhibitors, HIV-1 latency

## Abstract

Combined antiretroviral therapy (cART) is treatment with a combination of several antiretroviral drugs that block multiple stages in the virus replication cycle. An estimated 60% of the 38 million HIV-1 patients globally receive some form of cART. The benefits of cART for controlling HIV-1 replication, transmission, and infection rates have led to its universal recommendation. Implementation has caused a substantial reduction in morbidity and mortality of persons living with HIV-1/AIDS (PLWHA). More specifically, standard cART has provided controlled, undetectable levels of viremia, high treatment efficacy, reduction in pill burden, and an improved lifestyle in HIV-1 patients overall. However, HIV-1 patients living with AIDS (HPLA) generally show high viral loads upon cART interruption. Latently infected resting CD4+ T cells remain a major barrier to curing infected patients on long-term cART. There is a critical need for more effective compounds and therapies that not only potently reactivate latently infected cells, but also lead to the death of these reactivated cells. Efforts are ongoing to better control ongoing viral propagation, including the identification of appropriate animal models that best mimic HIV-1 pathogenesis, before proceeding with clinical trials. Limited toxicity profiles, improved drug penetration to certain tissues, and extended-release formulations are needed to cover gaps in existing HIV-1 treatment options. This review will cover past, current, and new cART strategies recently approved or in ongoing development.

## 1. Introduction

HIV-1 establishes a stably integrated, non-productive latent state of infection of individual cells, mainly long lived CD4+ T cells that are maintained by homeostatic proliferation [1,2]. This latent and stable reservoir is a primary barrier to HIV-1 eradication, despite the evident success of combined antiretroviral therapy (cART) [3]. cART effectively silences HIV-1 replication, but the persistence of latent reservoirs in the myeloid [4,5] and T cells of patients makes HIV-1 infection incurable [3,6,7,8]. HIV-1 can also replicate in brain microglial cells, which persist despite cART [9,10]. Honeycutt et al. [11] reported that integrated HIV-1 DNA is present is human bone marrow and spleen macrophages even after cART, and that mice with only human myeloid cells allow persistent infection in macrophages during cART in vivo.

The Department of Health and Human Services (DHHS) guidelines on antiretroviral agents for PLWHA encourages treatment for everyone who wants treatment and understands its importance [12]. The WHO has recommended that all PLWHA take cART regardless of their clinical status or CD4+ cell numbers, and the WHO supports promptly starting cART treatment of PLWHA, including administering cART on the same day as the diagnosis [13]. The DHHS guidelines suggest cART with two nucleoside reverse transcriptase inhibitors (NRTIs) in combination with a third active drug for treatment-naive HIV positive individuals with completely susceptible virus [14]. The third active drug of choice consists of an integrase strand transfer inhibitor (INSTI).

## 2. First AIDS Drug—AZT

The first AIDS drug, zidovudine (AZT, HIV-1 reverse transcriptase inhibitor (RTI [15])) was released almost five years after the discovery of HIV-1 [15,16]. Upon approval by the US Food and Drug Administration (FDA) in March 1987, AZT-based monotherapy provided the US public confidence that AIDS, considered a death sentence at the time, could be relatively controlled. The lack of major success by AZT (due to drug toxicity causing severe anemia and liver problems in HIV-1 patients [17,18,19]) was followed by a few additional drugs in the late eighties (Hivid (ddC, zalcitabine), Videx (ddl, didanosine), and Zerit (d4T, stavudine)). All failed to obtain long lasting control of viremia because all had similar mechanisms of action and targeted only one step of the virus replication cycle. An additional issue was the requirement for multiple daily doses, which created complex dosing schedules. A shift toward targeting HIV-1 protease and integrase led to the design of a novel class of antiviral drugs, namely PIs, including Invirase (saquinavir), and the first combined antiretroviral therapy in the 1990s [20,21].

## 3. Highly Active Antiretroviral Therapy—HAART

The impact of highly active antiretroviral therapy (HAART) was evident by the time of the 1996 International AIDS Conference in Vancouver, when HIV-1 mortality rates started to resemble general mortality rates and viral loads became undetectable, largely putting the disease into remission [22]. HAART [23], through the combination of NRTIs, NNRTIs, and PIs, thus, significantly changed the progression and outcome of infection with HIV-1 [24]. The success of HAART changed this originally fatal disease into a treatable chronic infection. For example, a 20-year-old HIV-1 patient on HAART could optimally live into their 50s. By the early 2000s, however, there were obstacles in the use of HAART. High pill burdens, inconvenient dosing, and long-term toxicities contributed to poor compliance and the emergence of drug-resistant virus in many patients [25,26,27]. For those patients in whom antiviral drug-resistance developed, treatment options become limited and more complicated regimens were necessary to prevent further disease progression. PIs caused insulin resistance, cardiac arrhythmias, unbalanced redistribution of body fat (lypodystrophy), and required three daily intakes. Lactic acidosis [28] and peripheral neuropathy were caused by the use of NRTIs. cART regimens generally consist of a “backbone” of two NRTIs and a “base” of either a PI or NNRTI [28].

## 4. Combined Antiretroviral Therapy—cART

By the mid-2000s, new generations of drugs such as Viread (Tenofovir disoproxil fumarate) showed better antiretroviral performance, safer and longer-lasting regimens, and far fewer side effects with only one daily intake [29]. With this development, HIV-1/AIDS further became a chronic and manageable disease. For example, a 20-year-old HIV-1 patient under cART could optimally live into their 70s [30]. By the late 2000s, the elucidation of HIV-1 entry steps offered new opportunities for therapeutic intervention [31,32,33,34]. Two entry inhibitors were licensed in the mid-to-late 2000s, the fusion inhibitor enfuvirtide (T-20) [35] and the small-molecule CCR5 antagonist Maraviroc (MVC) [31,36,37]. For HIV-1 patients resistant to conventional drugs from the NRTI and PI classes, the entry and integration inhibitors approved in 2007 effectively suppressed HIV-1 while offering additional therapeutic options. Entry inhibitors are particularly attractive antiretrovirals because, unlike conventional antiretrovirals, they target HIV-1 extracellularly, thereby sparing cells from both viral- and drug-induced toxicities. In the following sections, some of the current antiretroviral options (Table 1) will be discussed. Table 1 contains a list of the current leading drugs among different antiretroviral groups (Entry Inhibitors, Integrase Strand Transfer Inhibitors (INSTIs), Nucleoside Reverse Transcriptase Inhibitors (NRTIs), anti-CD4 Monoclonal Antibodies, Nucleoside Reverse Transcriptase Translocation Inhibitors (NRTTIs), Capsid Inhibitors (CAIs), Attachment Inhibitors that bind gp120 on HIV-1, and Latency-Reversing Agents (LRAs)) as well as their phase of current development.

## 5. gp120-Binding Proteins Inhibit HIV-1 Infection

Several HIV entry inhibitors that target HIV Env gp120 and gp41 are potent HIV inhibitors [38]. These include protein-based inhibitors such as soluble CD4 protein (sCD4) [39], eCD4-IgG [40], and antibodies against gp120, N6 [41], and m36.4 [42]. sCD4 targets the CD4-binding site (CD4bs) of gp120 and inhibits HIV entry and infection [39]. Schiavone et al., 2012 [43], designed a peptide mimotope of the HIV-1 gp120 bridging sheet. Their data validated the peptide mimotope approach as a promising tool to obtain an effective HIV-1 vaccine. Most recent studies suggest a potential to expand the previous strategy of combining a gp120-binding protein and a gp41-binding antibody for the treatment of HIV-1 infection [44]. Wang et al., 2021 [44] (originally identified a gp120-binding protein, mD1.22 as an inactivator of laboratory-adapted HIV-1) found that a gp41 N-terminal heptad repeat (NHR)-binding antibody D5 single-chain variable fragment (scFv) alone cannot inactivate HIV-1, even at high concentrations. However, a combination with D5scFv resulted in an enhanced inactivation activity of mD1.22 against divergent HIV-1 strains. These strains include primary HIV-1 isolates, T20- and AZT-resistant strains, HIV-1 laboratory-adapted strains, and LRA-reactivated virions. The authors [44], demonstrated that combining mD1.22 and D5 scFv gave synergistic inhibition of divergent HIV-1 strains.

## 6. Entry Inhibitors

T-20 and MVC are the first entry inhibitors licensed for patients with drug-resistant HIV-1, with MVC restricted to those infected with CCR5-tropic HIV-1 (R5 HIV-1) only. In addition, given that the oral administration of MVC offers high drug levels in cervicovaginal fluid, combinations of MVC and other CCR5 inhibitors have shown high effectiveness in preventing HIV-1 transmission. In addition, CCR5 antagonists prevent rejection of transplanted organs; therefore, MVC suppresses HIV-1 and prolongs organ survival for the growing number of HIV-1 patients with kidney or liver failure necessitating organ transplantation. Thus, MVC offers an important treatment option for HIV-1 patients with drug-resistant R5 HIV-1, who presently account for >40% of drug-resistance cases [45,46].

## 7. Integrase Strand Transfer Inhibitor (INSTI), Dolutegravir (DTG), and Nucleoside Reverse Transcriptase Inhibitor, Lamivudine

The most recent 2018 WHO recommendations list dolutegravir with an NRTI backbone as the preferred first-line ART regimen (WHO from review). This drug belongs to the INSTI class and has high tolerance to resistance. It is commonly recommended by the WHO as the first- and second-line agent for PLWHA [13] in recent triple drug cART, due to DTG’s better antiretroviral efficacy over raltegravir (RAL) [47,48]. In treatment-experienced patients, cART regimens based on once-daily DTG showed greater viral suppression when compared to twice-daily RAL (71% DTG versus 64% RAL) [47]. McAllister et al., 2017, showed that DTG with tenofovir disoproxil fumarate and FTC is an effective option for once daily post-exposure prophylaxis in men who have sex with men (MSM) [48]. In addition, DTG exhibits a higher barrier to resistance than RAL does. Additionally, it has a low interaction potential; therefore, there are no food restrictions [49,50,51]. Based on the most recent Guidelines for the Use of Antiretroviral Agents in Adults and Adolescents with HIV-1 [52], DTG has become a staple in combined and recommended regimens for most people with HIV-1; however, there are competing treatments such as bictegravir with emtricitabine and tenofovir alafenamide [53].

The first two-drug therapy (Juluca) was FDA approved in 2017 for certain HIV patients, followed by Dovato, which was FDA approved in 2019 for both treatment-experienced and treatment-naïve patients [54,55,56]. Dovato includes the new generation of integrase inhibitor (DTG) with the NRTI, lamivudine. The two-drug combination had fewer side effects than the standard triple drug therapy with the same effectiveness. Darunavir, the first non-INSTI antiretroviral treatment, was approved in 2011 as a single tablet [57]. In 2021, an injectable therapy, Cabenuva (cabotegravir and rilpivirine), was released [58]. Cabenuva is a new and complete prescription regimen and is used to treat HIV-1 infection in adults as a replacement for their current HIV-1 treatment when their healthcare provider determines that they meet certain requirements. Cabenuva is available as an injection that is administered once a month [58].

## 8. Available HIV-1 Treatment Options

### 8.1. Anti-CD4 Monoclonal Antibodies

Viral fusion and entry are primarily achieved by the gp120-gp41-CD4 complex undergoing multiple conformational changes. The first anti-CD4 monoclonal antibodies (mAbs) were introduced in the early 1990s. Ibalizumab is a humanized mAb that binds to the N-terminus of the second CD4 receptor’s domain. It has 10-fold higher antiretroviral activity while neutralizing most of the 116 HIV strains with 50% infection suppression [59]. Of the patients examined, 43% had a detectable viral load measuring below 50 copies/mL after 6 weeks of treatment [60]. Resistance to Ibalizumab via loss of glycosylation sites in the envelope V5 loop was evident during phase Ib clinical trials in 80% of the patients experiencing virologic failure. The outcomes of a phase III trial (optimized background regimen in MDR HIV-1) secured US regulatory approval for MDR HIV-1 treatment.

### 8.2. Nucleoside Reverse Transcriptase Translocation Inhibitors (NRTTI)

Unlike NRTIs, the nucleoside reverse transcriptase translocation inhibitors have dual mechanisms of action. In combination with a 3′-hydroxyl group, a 4′-ethynyl group inhibits translocation, resulting in chain termination [61,62]. A first-in-class NRTTI, Islatravir, was used as a monotherapy in Phase I clinical trials with treatment-naive HIV-1 patients, showing a mean 1.2 log10 decline in HIV-1 RNA copies daily [63]. Islatravir is an investigational NRTTI currently being evaluated for different frequencies and doses for HIV-1 treatment as both a single drug and in combination with other antiretroviral drugs at the most recent Conference on Retroviruses and Opportunistic Infections, CROI 2021 [64]. The study demonstrates that the implant showed active drug concentrations above the pre-specified PK threshold at 12 weeks. The tested Islatravir doses in the study were 48, 52, and 56 mg. The implant was projected to provide drug concentrations likely above the threshold for one year at the highest dose. Based on these findings, Merck plans to initiate a Phase 2 trial. They will explore the potential of a subdermal implant containing Islatravir as a long-acting pre-exposure prophylaxis (PrEP) option for up to a year. There is also an ongoing phase IIa randomized control trial to evaluate oral Islatravir as a PrEP option once monthly [65]. In addition, there is a double-blind randomized control study using Islatravir implants in two different doses, which is slightly different from the Merck study (54 and 62 mg), conducted with 16 healthy individuals for the purposes of evaluating the pharmacokinetics profile and tolerability of the Islatravir implants. This study showed that at 12 weeks, both implants reached above the target PK thresholds in one year. Adverse events were mild to moderate, with no clinical differences between the treatment and placebo experimental groups. The 62-milligram implant group reported higher pain than the 54-milligram implant group did [66].

### 8.3. Capsid Inhibitors

HIV-1 infection depends on the orderly formation and dissolution of the viral capsid. This dual role of the capsid protein in viral maturation and post-entry uncoating makes it a very attractive target of opportunity for antiretroviral therapies. The pre-integration complex (containing the viral DNA in the capsid core) is transported through the cell cytoplasm to the nucleus. The capsid protein is cleaved from Gag polyprotein precursors to form the capsid core during virion maturation. The capsid core undergoes an uncoating process, during which capsid hexamers disassemble after fusion between the virus membrane and the target cell membrane. The released reverse transcription complex is then transported to the cell nucleus. Numerous capsid protein inhibitors (CAIs) have been reported to block uncoating and HIV-1 infection. These include CAP-1, PF74, BI compounds, peptide inhibitors, Coumermycin A1, C1, Ebselen, anti-capsid antibodies, and GS-CA1. CAIs were first introduced at the Conference on Retroviruses and Opportunistic Infections, CROI 2017, as a first-in-class picomolar capsid inhibitor [67]. In 2017, GS-CA1, a highly potent capsid inhibitor, was described as holding promise for clinical development [58]. Although no CAI is currently approved for clinical use, GS-CA1 is a potent CAI that holds much potential for therapeutic development against HIV-1 [68]. CAI’s main advantages are a high barrier to resistance and long-acting potential due to low predicted hepatic metabolic clearance, based on cryopreserved hepatocyte models. Good water solubility and the longer half-life of compound GS-6207 (which binds to the linker connecting the N-terminus and C-terminus domains that form the capsid protein) offered the option of monthly subcutaneous dosing. Compound GS-6207 is currently being successfully tested in Phase Ib of randomized controlled trials to estimate its antiretroviral efficacy in treatment-experienced and treatment-naive HIV-1 patients [69]. This study, thus far, shows a very impressive mean of a 1.8 to 2.2 log10 decline in HIV-1 RNA copies at day 10 after a single subcutaneous dose of the compound. There were no indications of grade three or four adverse events requiring discontinuation [70]. In summary, capsid-targeting drugs are predicted to have high barriers to HIV-1 resistance.

### 8.4. Latency-Reversing Agents (LRAs)

As mentioned earlier, cART cannot eradicate HIV-1 due to the latent viral DNA present in cell reservoirs. It is also imperative to ensure minimal depletion of non-infected cells. LRAs are considered viable options to cure HIV-1 by destabilizing latency and causing immune clearance of infected cells and could lead to treatment-free remission. HIV-1 latency itself depends in large part on the silencing environment of the infected host cell, which can be chemically altered. The original concept, known as “shock and kill” [3,71], depends upon HIV-1 being functionally reactivated by LRAs in target cells, allowing them to be eliminated by immune effectors such as cytotoxic T lymphocytes or by virus-induced cytopathic damage (Figure 1).

“Shock and kill” is intended to reverse proviral quiescence by inducing viral transcription and allowing a mixture of antiretroviral therapy, host immune clearance, and HIV-cytolysis to remove the remaining and latently infected cells, leading to a full virologic cure. Timmons et al. [72] showed that for diverse LRAs, latency reversal in model systems involves the heat shock factor 1 (HSF1)-mediated stress pathway. The small-molecule inhibition of HSF1 attenuated HIV-1 latency by histone deacetylase inhibitors, protein kinase C agonists, and proteasome inhibitors without interfering with the known mechanism of action of these LRAs. They demonstrated that in vitro models of latency have higher levels of the P-TEFb subunit cyclin T1 than primary cells do, which may explain why many LRAs are functional in model systems but relatively ineffective in primary cells. The failure of LRA in clinical trials was evidenced by a lack of reduction in the size of the reservoirs after LRA implementation. The majority of LRAs identified to date have been relatively ineffective despite activity in model systems.

Alternative therapies such as the HIV-1 transcription-inhibiting “block and lock” strategy to drive the pro-virus into a state of deep latency (utilizing latency promoting agents (LPAs) targeting either HIV or host-specific mechanisms) are, therefore, being considered [73]. Furthermore, the “block and lock” has arguments supporting its use over the “shock and kill” hypothesis due to less adverse side effects. More importantly, the concern that “shock and kill” might never completely eradicate the proviral reservoir reduces its viability for common use. Furthermore, numerous LRAs lack the “kill” aspect; therefore, the “shock and kill” therapy should be supplemented with existing immunomodulatory options. Additional novel strategies to manipulate the latent reservoirs, such as “block and lock”, should be explored further. In addition, there is a need for approaches that induce the death of latently infected cells through the apoptosis and inhibition of cellular pathways critical for cell survival. Those pathways are often hijacked by HIV-1 proteins. Given the advances in the commercial development of compounds that induce apoptosis in cancer chemotherapy, these agents could move rapidly into HIV-1 latency clinical trials, either alone or in combination with existing LRAs in order to target and eliminate latent HIV-1 infection. The development of highly specific LRAs is warranted to contribute to the eradication of HIV-1 [74].

## 9. Remaining Obstacles

Despite enormous successes, the remaining obstacles hindering current cART options are viral persistence, drug toxicities, emergence of drug resistance against existing antiretroviral regimens, and side effects. For example, current cART drug combinations also suffer from increasing pre-existent drug resistance in treated patients and transmission of those resistant variants to others. Drug resistance affects the treatment outcome, and undetectable drug resistant mutants occur even in suppressed patients [75,76], which is why new combinations of antiretrovirals are needed for better long-term safety, tolerability, adherence to cART regimens, and barriers to resistance. Second, multiple studies have demonstrated that activation of viral transcription alone is not sufficient to induce cell death, and some LRAs may have the unwanted effect of counter-acting cell death by promoting cell survival [77]. In addition, clinical trials with LRAs have demonstrated that activation of viral gene expression is possible in vivo, but there is very limited or no clearance at all in the reactivated cells [78,79,80,81]. Efforts are being made to develop combinations of two or three synergizing drugs to give the highest possible potency with the lowest side effects, preferably with lower costs. This includes examining the interactions between LRAs and other drugs to determine potential synergy, antagonism, or toxicity.

### 9.1. Viral Persistence

Obstacles such as viral persistence are best considered in the context of viral replication in vivo. Viral replication in HIV-1 patients is largely the consequence of a dynamic process involving continuous rounds of de novo HIV-1 infection of and replication in activated CD4+ T cells with a rapid turnover of free virus and virus-infected cells. This process is significantly, but not completely, disturbed by effective cART. After a few months of cART therapy, plasma viral loads become undetectable in most patients. Functional assay-based laboratory analysis demonstrating evidence of decreased viral levels initially suggested that eradication of the HIV-1 infection might be achievable. Despite this evidence, there are several potential cellular and anatomical reservoirs for HIV-1 persistence that may contribute to the long-term latency of HIV-1. For example, latently infected resting memory CD4+ T cells carrying integrated HIV-1 DNA, HIV-1 infected cells in the gut-associated lymphoid tissue (GALT), central nervous system (CNS), and the male urogenital tract are active and identified locations for HIV-1 reservoirs. The half-life of resting CD4+ T cells is extremely long (3.7 yrs). That means that the eradication of this reservoir would require over 60 years of cART treatment, which is impractical as a viable eradication strategy. Latently infected resting CD4+ T cells provide a mechanism for life-long persistence of replication-competent forms of HIV-1, rendering hopes of virus eradication with current antiretroviral regimens unrealistic. The extraordinary stability of the reservoir may also reflect gradual reseeding by a very low level of ongoing viral replication and/or mechanisms that contribute to the intrinsic stability of the memory T cell compartment.

### 9.2. Side Effects

Obstacles such as substantial long-term toxicities and the side effects of current cART regimens require developing novel approaches to eradicating latent reservoirs. Solutions are urgently needed and might be addressed by the use of non-invasive CCR5 targeting drugs to intensify standard cART options. The cART refractory latent reservoirs contributing to the rapid virus rebound that occurs when latent cells become reactivated [21] prevent cART from completely eradicating HIV-1.

Efforts to develop alternative strategies have been further stimulated by the apparent cure of the “*Berlin patient*” [82] by a bone marrow transplant from a donor homozygous for a mutant defective CCR5 gene (CCR5 ∆32), which confers resistance to R5 HIV-1 [83,84]. This confirmed CCR5 as an appealing target for antiviral drugs and suggested that adding CCR5 targeting drugs to cART could increase antiviral efficacy. CCR5 is also an ideal antiviral target due to its relatively disposable role in the human immune system [85,86], suggesting that CCR5 targeting drugs may have low toxicity. By increasing antiviral efficacy, it might be possible to deplete viral reservoirs in gut-associated lymphoid tissue (GALT) and address the issue of inadequate cART distribution into this compartment. CCR5 signaling may facilitate trafficking T cells to areas of inflammation [87] and blocking such trafficking could further reduce viral spread and active replication.

### 9.3. Poor Drug Penetration

cART is not able to suppress HIV-1 replication fully due to poor drug penetration into certain tissues such as the central nervous system [88]. Existing cART drug combinations are effective, but also suffer from increasing pre-existing drug resistance in treated patients and the transmission of those resistant variants to others. Drug resistance can affect treatment outcome and undetectable resistant mutants occur even in successfully suppressed patients [75,76]. New combinations of antiretrovirals are clearly needed for better long-term safety, tolerability, adherence to cART regimens, and barriers to resistance.

Gastrointestinal tract CCR5+ CD4+ T cells (GALT) are selectively infected and depleted during acute HIV-1 infection. GALT T cell depletion and activation persist despite cART. Persistent infection targets include long-lived CD4+ T cell subpopulations and myeloid lineage cells [89,90]. Provirus in these cells can be transcriptionally inactive for long periods, enabling the virus to evade cART drugs and immune responses [82,91]. It is difficult to reach infected cells harbored in some solid tissues, such as the central nervous system and GALT, with existing antiviral drugs. These are challenging obstacles for virologic cures [92,93,94,95]; therefore, identifying and characterizing these sites of viral persistence is a critical step toward a cure. New strategies are needed because neither the combination of latency reversing agents [89] and cART, nor host immune responses [90,91] appear to effectively reduce the pool of latently infected cells in GALT. Drug potency and specificity could be potentially enhanced using drug delivery systems such as nanoparticles coated with specific antibodies targeting CD4 or a latency marker, such as the recently described CD32a [96].

## 10. Summary

Since HIV-1 has become a chronic illness managed with various cART options, the number of patients over the age of 50 years keeps increasing with patients functioning and living successfully. Commitment to cART treatment is essential for long-term success. In addition, many different antiretroviral drugs successfully target different stages of the viral replication cycle with new combined therapeutics with lower side effects. The introduction of a lower cost cART regimen consisting of a single daily pill, injectable medications, and drug-eluting implants may significantly increase the management of HIV-1/AIDS and expand options for the utility of PreP medications for HIV-1. Altogether, this should improve the management of HIV-1/AIDS. Continuous and ongoing efforts are warranted to address eliminating reservoirs causing HIV-1 latency, and we emphasize the importance of investing research efforts into an HIV-1 cure. Identifying novel compounds that can be combined to both induce reactivation and death of HIV-1 latently infected cells are critical in this effort, as this strategy is not dependent upon an effective immune response or understanding immune escape in HIV-1 latently infected human cells.

## Figures and Tables

**Figure 1 pharmaceutics-13-01798-f001:**
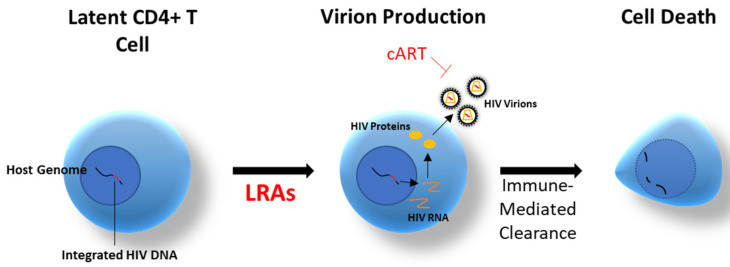
The “Shock and Kill” Strategy uses Latency Reversing Agents for Immune-Mediated Clearance.

**Table 1 pharmaceutics-13-01798-t001:** Leading drug options among anti-HIV groups of drugs. Information was retrieved from the NIH US National Library of Medicine (https://www.clinicaltrials.gov/ct2/home accessed on 4 June 2021).

Antiretroviral Drug Group	Leading Drug Option	Phase of Current Development	FDA Approval Status	Route of Administration in HIV-1 Patients
**Entry Inhibitors**	Enfuvirtide (T-20)	IV Completed	Yes	Oral
	Maraviroc	IV Completed	Yes	Oral
	Zinc-Finger Nuclease	II Completed	No	Infusion
**Attachment Inhibitors**	Fostemsavir (Rukobia)	III Active	Yes	Oral
**Anti-CD4 Monoclonal Antibodies**	Ibalizumab	III Completed	Yes	Infusion
**Nucleoside Reverse Transcriptase Translocation Inhibitors (NRTTI)**	Islatravir	I Completed, II Active	No	Oral
**Integrase Strand Transfer Inhibitors (INSTI)**	Dolutegravir	IV Completed	Yes	Oral
**Capsid Inhibitors**	GS-6207	Ib Completed, II/III Active	No	Oral/Subcutaneous
**Latency-Reversing Agents (LRAs)**	Romidepsin	II Completed	No	Infusion

## Data Availability

Not Applicable.

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
