# Peer review of "Past HIV-1 Medications and the Current Status of Combined Antiretroviral Therapy Options for HIV-1 Patients"

_pharmaceutics, 2021, doi:10.3390/pharmaceutics13111798_

Round 1
Reviewer 1 Report
Comments and Suggestions for Authors
Minor comments:
- “ The DHHS guidelines suggest cART with two nucleoside reverse transcriptase inhibitors (NRTIs) in combination with a third active drug for treatment-naive HIV-positive individuals with the completely susceptible virus [14]. The third active drug of choice can be a non-nucleoside reverse transcriptase inhibitor (NNRTI), a protease inhibitor, or an integrase strand transfer inhibitor (INSTI). “ P2, L48~52.
Please revised the previous comment “ DHHS guideline for the use of ARV agents in PLWHA is last updated on December 18, 2019, and the new vision is updated on June 3, 2021.”
The authors can link to the website as below:
https://clinicalinfo.hiv.gov/en/guidelines/adult-and-adolescent-arv/what-start-initial-combination-regimens-antiretroviral-naive?view=full
- Please revise the “ protease inhibitor and
nonnucleoside reverse transcriptase inhibitor"
abbreviation in the main text. - The paragraph “The third active drug of choice can be a non-nucleoside reverse transcriptase inhibitor (NNRTI), a protease inhibitor, or an integrase strand transfer inhibitor (INSTI). “It is unclear? The INSTIs are the main third active agent now as the initial combination regimens for the antiretroviral-naive patient
- Please revise the number “ 52. Guidelines for the Use of Antiretroviral Agents in Adults and Adolescents with HIV. ” Please sea P4, L 152
5. P8L340: please unify the references according to the author's guidelines.

Author Response
REVIEWER 1 Minor comments:
- “ The DHHS guidelines suggest cART with two nucleoside reverse transcriptase inhibitors (NRTIs) in combination with a third active drug for treatment-naive HIV-positive individuals with the completely susceptible virus [14]. The third active drug of choice can be a non-nucleoside reverse transcriptase inhibitor (NNRTI), a protease inhibitor, or an integrase strand transfer inhibitor (INSTI). “ P2, L48~52.
Please revised the previous comment “ DHHS guideline for the use of ARV agents in PLWHA is last updated on December 18, 2019, and the new vision is updated on June 3, 2021.”
The authors can link to the website as below:
https://clinicalinfo.hiv.gov/en/guidelines/adult-and-adolescent-arv/what-start-initial-combination-regimens-antiretroviral-naive?view=full
Thank you for your comment. We have corrected it to PLWHA throughout the document (highlighted), but we prefer to keep cART as is because combined antiretroviral therapy is the common term in the HIV field.
- Please revise the “ protease inhibitor and
nonnucleoside reverse transcriptase inhibitor"
abbreviation in the main text.
We’ve changed the incorrect abbreviations on Line 72. See highlighted section.
- The paragraph “The third active drug of choice can be a non-nucleoside reverse transcriptase inhibitor (NNRTI), a protease inhibitor, or an integrase strand transfer inhibitor (INSTI). “It is unclear? The INSTIs are the main third active agent now as the initial combination regimens for the antiretroviral-naive patient
Thank you for your comment. It has been corrected in Line 50-52. The highlighted part of the sentence.
- Please revise the number “ 52. Guidelines for the Use of Antiretroviral Agents in Adults and Adolescents with HIV. ” Please sea P4, L 152
The reference number 52 is correctly listed as the Guidelines for the Use of Antiretroviral Agents in Adults and Adolescents with HIV. We’ve updated the reference in the reference list to give more information about the reference.
- P8L340: please unify the references according to the author's guidelines.
The references 93 and 94 in line 340 are listed according to the Journal’s guidelines for the Authors

Reviewer 2 Report
The authors addressed every highlighted points.
Author Response
We appreciate the time that the Rev 2 took to see our manuscript again. He/she did not have any comments. Thank you.